# The Impact of Co-Infections for Human Gammaherpesvirus Infection and Associated Pathologies

**DOI:** 10.3390/ijms241713066

**Published:** 2023-08-22

**Authors:** Prishanta Chinna, Katrin Bratl, Humaira Lambarey, Melissa J. Blumenthal, Georgia Schäfer

**Affiliations:** 1International Centre for Genetic Engineering and Biotechnology (ICGEB), Cape Town 7925, South Africa; pllpri021@myuct.ac.za (P.C.); brtkat011@myuct.ac.za (K.B.); lmbhum001@myuct.ac.za (H.L.); melissa.blumenthal@icgeb.org (M.J.B.); 2Institute of Infectious Disease and Molecular Medicine (IDM), Faculty of Health Sciences, University of Cape Town, Cape Town 7925, South Africa; 3Division of Medical Biochemistry and Structural Biology, Department of Integrative Biomedical Sciences, Faculty of Health Sciences, University of Cape Town, Cape Town 7925, South Africa

**Keywords:** Epstein-Barr virus (EBV or HHV-4), Kaposi’s sarcoma-associated herpesvirus (KSHV or HHV-8), human immunodeficiency virus (HIV), *Mycobacterium tuberculosis* (Mtb), *Plasmodium falciparum*, severe acute respiratory syndrome coronavirus 2 (SARS-CoV-2), malaria, COVID-19, Sub-Saharan Africa (SSA)

## Abstract

The two oncogenic human gammaherpesviruses Epstein-Barr virus (EBV) and Kaposi’s sarcoma-associated herpesvirus (KSHV) cause significant disease burden, particularly in immunosuppressed individuals. Both viruses display latent and lytic phases of their life cycle with different outcomes for their associated pathologies. The high prevalence of infectious diseases in Sub-Saharan Africa (SSA), particularly HIV/AIDS, tuberculosis, malaria, and more recently, COVID-19, as well as their associated inflammatory responses, could potentially impact either virus’ infectious course. However, acute or lytically active EBV and/or KSHV infections often present with symptoms mimicking these predominant diseases leading to misdiagnosis or underdiagnosis of oncogenic herpesvirus-associated pathologies. EBV and/or KSHV infections are generally acquired early in life and remain latent until lytic reactivation is triggered by various stimuli. This review summarizes known associations between infectious agents prevalent in SSA and underlying EBV and/or KSHV infection. While presenting an overview of both viruses’ biphasic life cycles, this review aims to highlight the importance of co-infections in the correct identification of risk factors for and diagnoses of EBV- and/or KSHV-associated pathologies, particularly in SSA, where both oncogenic herpesviruses as well as other infectious agents are highly pervasive and can lead to substantial morbidity and mortality.

## 1. Introduction 

Human herpesviruses are large, enveloped, linear, double-stranded (ds) DNA viruses with genomes that range between 125 and 240 kb in length. With few exceptions, they occur at a high prevalence in the human population and are associated with a wide spectrum of clinical manifestations, ranging from asymptomatic infection to severe disease. Herpesviruses are characterized by their ability to establish life-long latent infections, maintaining their genome in a persistent circular form. While the initial lytic cycle of primary infection leads to the generation of large numbers of virions, latency is characterized by the absence of virus production, where only a limited set of viral genes is expressed. Latency can be interrupted by periods of intermittent lytic reactivation where infectious progeny is produced and shed [1,2].

There are currently eight known human herpesviruses with distinct host cell specificities and clinical characteristics, which are classified into three subfamilies: alpha-, beta-, and gammaherpesvirinae. The alphaherpesvirus group comprises herpes simplex virus type 1 (HSV-1 or HHV-1), herpes simplex virus type 2 (HSV-2 or HHV-2), and Varicella-Zoster virus (VZV or HHV-3), while human cytomegalovirus (HCMV or HHV-5), human herpesvirus-6 (HHV-6), and human herpesvirus-7 (HHV-7) belong to the betaherpesvirus group. The two oncogenic herpesviruses Epstein-Barr virus (EBV or HHV-4) and Kaposi’s sarcoma-associated herpesvirus (KSHV or HHV-8) are members of the gammaherpesvirus group. Importantly, the alpha- and betaherpesvirus subfamilies mainly attribute their pathogenic potential to viral replication and lytic reactivation, while life-long latent infection typically has minimal impact upon the host. However, both latent and lytic gene products contribute to malignant transformation and associated pathologies caused by the gammaherpesvirus subfamily [3].

Various environmental factors can upset the delicate balance between herpesvirus latency and lytic reactivation, which can lead to disease onset. Most of these factors have been identified by in vitro experiments and include chemical stressors that impact specific intracellular signaling cascades and/or epigenetic regulation. Clinically, triggers of lytic reactivation also include hypoxia, inflammation (including oxidative and nitrative stress), DNA damage due to UV exposure, or high levels of autonomic nervous system (ANS) activity [4,5,6]. Importantly, various co-infections have been reported to lead to herpesvirus reactivation, which has been, for example, observed on herpesvirus-infected B cells when the infected B cell responds to unrelated infections [7].

Pathologies associated with the two oncogenic gammaherpesviruses EBV and KSHV are particularly prevalent in human immunodeficiency virus (HIV)-infected individuals, with Kaposi’s sarcoma (KS) being the most common acquired immunodeficiency syndrome (AIDS)-related malignancy worldwide. Due to the high prevalence of HIV/AIDS and other infectious diseases in Sub-Saharan Africa (SSA), such as tuberculosis, malaria, and more recently, COVID-19, special focus must be placed on co-infections implicated in herpesvirus-associated pathologies, particularly those with oncogenic potential. Although both EBV and KSHV are highly prevalent in SSA, the influence of co-infections on their associated diseases is not well understood. This review will therefore focus on known and emerging co-infections in SSA and their impact on the lytic reactivation of the two human oncogenic herpesviruses EBV and KSHV.

## 2. Epidemiology of the Oncogenic Gammaherpesviruses EBV and KSHV

### 2.1. EBV

EBV infects almost 90% of the global population, but primary EBV infection rates differ according to geographic region, age, and socioeconomic status, with higher rates of infection in developing countries [8,9,10]. EBV consists of two different genotypes, type 1 and 2, or A and B, respectively, based on the differences in the EBV-associated genes, EBNA-2 and EBNA-3 [11,12]. Type 1 EBV has been shown to have a worldwide distribution, whereas type 2 EBV is particularly prevalent in SSA. These strains differ in their ability to induce growth transformation, with type 1 transforming B cells into lymphoblastoid cell lines more efficiently than type 2 [11,13,14].

EBV is mainly transmitted through infected saliva, primarily in childhood; however, it can also spread via blood transfusions and organ transplants [15,16,17,18]. It has been suggested that almost all children in underdeveloped countries are EBV-seropositive by the age of 6 years old compared to developed countries where seropositivity peaks at 2–4 and 14–18 years of age and thereafter increases with age [19,20,21].

While primary EBV infection is asymptomatic in most cases, it can lead to infectious mononucleosis (IM) (a self-limiting disease characterized by lymph node inflammation in the neck region [15]). Studies have established that most primary EBV infections in children under 2 years of age are asymptomatic; however, some children between 10 and 48 months old can display symptoms such as fever, tonsillar pharyngitis, prominent cervical lymphadenopathy, and respiratory symptoms during IM [22,23,24] with a large difference in heterophile antibody responses between infants less than 2 years of age (27.3%), children between 25 and 48 months old (76.2%), and young adults (96%) [23].

Long-term EBV infection is further associated with several malignancies such as Burkitt’s lymphoma (BL), Hodgkin’s lymphoma (HL), T-cell lymphomas, nasopharyngeal carcinoma, gastric cancer, and post-transplant lymphoproliferative disease, particularly in immunosuppressed and immunocompromised individuals, especially patients with HIV/AIDS, transplant recipients, or those undergoing chemotherapy [15,25,26,27,28,29,30,31,32]. In these patients, EBV-infected cells are able to evade the host’s immune system, enable metabolic reprogramming, modulate apoptosis, and encourage cancer metastasis and tumor proliferation [25].

In SSA, EBV co-infections with HIV, *Mycobacterium tuberculosis* (Mtb), *Plasmodium* sp., and other viruses, such as KSHV, HCMV, and HSV, are particularly relevant to address. Symptoms of EBV-associated pathologies are often disguised by these prevalent diseases and left undiagnosed, potentially exacerbating EBV viral infection in the host [33,34,35,36,37,38].

EBV-associated cancers account for approximately 1.5% of all human cancers globally [39] and are responsible for 137,900–208,700 cancer-related deaths annually [40]. The fact that EBV was initially discovered in African BL cell cultures is evidence of the disease’s high impact in this region, especially with the lack of efficient healthcare infrastructure and virus control strategies [40,41]. Endemic BL has been shown to have a particularly high impact in SSA affecting 80% of all pediatric patients presenting with hematologic malignancies and is further exacerbated by the high incidences of *Plasmodium* sp. and HIV infection in SSA countries [42].

### 2.2. KSHV

Unlike EBV, KSHV infection is not ubiquitous, and its prevalence varies geographically with seropositivity rates of less than 10% in the U.S. and Europe, moderate rates of 20–30% in Mediterranean countries, and high rates of 30–50% in SSA [43,44,45,46]. Based on sequence variations of the hypervariable Open Reading Frame (ORF)-K1 gene, KSHV can be subtyped into the overarching groups A, B, C, D, E, and F [47], with subtype A commonly observed in Northern Europe and America and subtype B being more predominant on the African continent and being considered the oldest existing strain [48]. There are several routes of KSHV transmission, such as through blood transfusions [49] or sexual contact [50]; however, similar to EBV, transmission via saliva during early childhood is considered the main route of transmission in endemic areas such as SSA [43,51,52,53].

KSHV is the causative agent of KS, primary effusion lymphoma (PEL), multicentric Castleman disease (MCD), and KSHV-associated cytokine syndrome (KICS), which primarily develop in HIV-infected individuals [54,55,56]. PEL, MCD, and KICS are rare but likely underreported KSHV-associated pathologies [57], whereas KS is considered the most common AIDS-related malignancy worldwide. Although KS is a relatively rare cancer on a global level, it is epidemic in SSA where both HIV and KSHV prevalence are exceptionally high. GLOBOCAN reported KS incidence rates of 8.5 (males) and 4.7 (females) per 100,000 in Southern Africa compared to incidences below 0.5 per 100,000 for Western European countries or Asia, in 2020 [58]. Although advances in KS management of HIV-infected individuals were made by the introduction of antiretroviral therapy (ART), over 34,000 new cases and over 15,000 deaths were still reported in 2020 with KS being the leading cause of death in Mozambique and Uganda [58].

While HIV-related immune suppression, as defined by low CD4 counts, is considered the most critical mechanism promoting pathogenesis [59], other KSHV-associated malignancies such as MCD display preserved counts [60], highlighting the heterogeneity of KSHV-dependent pathologies. In addition to HIV, co-infection such as with Mtb, *Plasmodium* sp., HCMV, and HSV is highly common in SSA. The high burden of tuberculosis (TB), for instance, often leads to overdiagnoses of TB, and due to similar clinical presentation of lytic KSHV infection (such as fever, night sweats, inflammation and respiratory symptoms), underdiagnosis of KSHV-associated diseases [46].

Important to note, and different from EBV-associated malignancies, is that both lytic and latent gene products are involved in KSHV-associated tumorigenesis and are distinct between different KSHV-associated malignancies. While KS and PEL are generally characterized by latent gene products [61,62], lytic gene products leading to inflammatory symptoms are predominant in MCD and KICS, with fatal outcomes when untreated [55,63].

## 3. The Life Cycle of Oncogenic Gammaherpesviruses

### 3.1. The EBV Life Cycle and the Contribution of Key Latent and Lytic Gene Products

Both phases of the EBV life cycle play important roles in viral pathogenesis and the development of EBV-associated diseases [64]. However, latently infected B cells are predominantly associated with EBV-associated malignancies due to the disruption of cell growth, signal transduction mechanisms, and transcription control by a restricted set of latent genes produced [65]. EBV causes primary infection by replicating in the mucosal epithelium, thereby initially infecting oropharyngeal mucosa cells prior to infection of resting B lymphocytes in the underlying secondary lymphoid tissues [66]. The EBV virions generated and released from the epithelial cells favor B cells due to the presence of particular envelope glycoproteins and vice versa, thereby also allowing the virions to particularly infect naïve B lymphocytes [67,68].

The transformation of naïve B lymphocytes into EBV-infected memory B lymphocytes to establish latency is categorized into different latency stages/programs (0–III) depending on the expression of specific EBV-produced proteins [69]. EBV-infected B cells initially proliferate into activated lymphoblast cells and migrate to the germinal center (GC) of the lymph node follicle establishing the latency III program [69], resulting in the production of EBV nuclear antigens (EBNA)-1, -2, -3A, -3B, -3C, and leader protein (-LP) and latency membrane proteins (LMP)-1, -2A, and -2B, which regulate the growth of EBV [69,70,71]. EBNA-1, one of the most important latency proteins, shares structural homologies with the KSHV key latency protein latency-associated nuclear antigen (LANA, see Section 3.2) and is expressed during all latency program stages and during lytic replication [72]. In the GC, the latency I program is established with the production of EBNA-1 only [73,74,75] and the latency II program with EBNA-1, EBV-encoded small RNAs (EBERs), viral microRNAs (v-miRs), BamHI fragment A rightward transcripts (BARTs, i.e., transcripts of the viral genome), LMP-1, and LMP-2A [76,77]. These viral proteins and RNAs contribute to viral immune evasion, immunomodulation, prevention of apoptosis, limitation of viral replication, and survival by limiting interferon (IFN) signaling [78,79,80,81].

EBV-infected memory B lymphocytes in latency program 0 establish latency by persisting in a quiescent state with limited replication of only parts of the viral genome and limited expression of viral proteins [82,83]. The viral genome exists in the nucleus as a chromatinized, covalently closed, circular genome and is able to replicate once per cell cycle driven by the host’s DNA polymerase [84], remaining undetected in the host for years until reactivated. EBNA-1 is a sequence-specific multifunctional DNA-binding protein responsible for EBV episomal stability, persistence, and maintenance during latency 0 due to its anti-apoptotic properties and its ability to act as a transcriptional transactivator such as for LMP-1 [85,86,87]. EBNA-1 tethers the viral episome to the host genome and leads to subsequent B-cell immortalization [86].

The ability to maintain latency further relies on epigenetic mechanisms such as DNA methylation, which facilitates transcriptional silencing of the immediate early genes, BZLF1 (ZEBRA/Zta) and BRLF1 (Rta) [88,89]. The suppressive histone modification H3K27me3, as well as the presence of type II histone deacetylating complexes (HDACs) at the promoter region, further contributes to silencing BZLF1 (Zta) transcription [90,91].

Interestingly, reactive oxygen species (ROS) production by EBV was found to be required to immortalize B cells with a role in signal transduction and activation of transcription 3 (STAT3) phosphorylation; also, LMP-1 was shown to depend on ROS production, aiding its crucial role in the survival of latently infected cells [92,93].

During any of the latency stages, the memory B lymphocytes can differentiate into plasma cells and cause reactivation of the EBV infection with the production of new viral particles, which can be shed into the saliva [64]. Lytic reactivation of EBV can be triggered by many different factors, including psychological stress, immunosuppression, immunodeficiency, and the presence of foreign antigens due to new infections, which can lead to uncontrolled replication of EBV, potentially resulting in lymphoproliferative diseases [94,95]. During lytic reactivation of EBV, viral gene expression occurs in a sequential and regulated manner and is governed by the production and upregulation of the immediate early genes BZLF1 (ZEBRA/Zta) and BRLF1 (Rta), which activate a cascade of gene expression and signaling pathways [96]. The immediate early genes induce the expression of early genes responsible for viral DNA replication, such as BALF5, BALF2, BMRF1, and BSLF1, as well as late genes BCRF1 (vIL-10), BcLF1, and BNRF1, responsible for protecting the virus from the host immune system and for viral particle formation, respectively [97].

Activation of transcription factors via phosphorylation, as well as activation of cellular kinases such as JNK, MAPK/p38, ERK, PKC, PKD, and/or PI3K signaling/AKT, is important for reactivation and oncogenesis [95,98]. LMP-1 plays a critical role in cell proliferation, anti-apoptosis, transformation, metastasis, and invasion through NF-κB, PI3K, JAK/STAT-mediated, and MAPK-associated signaling activity [99]. EBNA-1 and LMP-2A are also associated with the inhibition of TGFβ1-induced apoptosis [99]. Conversely to its role in latency, ROS generation in EBV-transformed cell lines, induced via the interaction of the JNK and p38-MAPK signaling pathways, results in cellular apoptosis [100].

During lytic reactivation, cytokines such as interleukin (IL)-6, IL-8, IL-10, IL-13, and VEGF are upregulated by BZLF1 (Zta), increasing inflammation and potentially cellular proliferation [101]. The loss of control of any suppressive mechanism of the immediate early genes, BZLF1 (Zta) and BRLF1 (Rta), are fundamental for reactivation and transcriptional regulation of EBV [95,98].

### 3.2. The KSHV Life Cycle and the Contribution of Key Latent and Lytic Gene Products

Displaying broad cellular tropism, KSHV can infect various cell types in vivo, including peripheral B cells, monocytes, keratinocytes, endothelial cells, epithelial cells, and macrophages [102,103]. As infection mainly occurs upon oral exposure to infected saliva, cells that are associated with the mucosa (e.g., mucosa-associated lymphocytes, macrophages, and/or epithelial cells) are targeted first during primary infection. It is assumed that KSHV migrates within dendritic cells to lymphoid organs, from where it further infects B and T cells [104,105].

Latent and lytic infection phases have distinct viral gene expression patterns [106,107,108], which unfold differently among various cell types and are characteristic for each KSHV-associated disease; however, gene products of both phases are involved in tumorigenesis [61,62,109]. Importantly and unique among herpesviruses, the pool of latent cells is maintained by lytically active cells, emphasizing how both viral cycles play their role in pathogenesis [6]. B cells and monocytes are considered the major reservoir of latent KSHV, and in KS as well as PEL, gene expression is predominantly latent. Whereas 99% of tumor cells (primarily of endothelial origin) in KS lesions are latently KSHV-infected, high rates of proliferation occur in spindle cells, accounting for only 1% of tumor cells, and have been reported to contribute to higher motility, proliferation, and sustaining angiogenesis [103,110,111,112]. In MCD, both lytic and latent genes are expressed in B cells surrounding the GCs of lymph nodes [55].

Latency is a controlled but reversible state where the KSHV genome persists as a highly ordered chromatin structure within the host, and lytic gene expression is suppressed [102,103]. The major latency locus of KSHV consists of ORF K12 (Kaposins A–C), ORF71 (v-FLIP), ORF72 (v-CYC), and ORF73 (LANA), as well as 12 pre-miRNA sequences [113,114,115,116]. Structurally homologous to the EBV key latency protein EBNA-1 (see Section 3.1), LANA is the major regulator of latency and is considered a multifunctional oncoprotein. Through its interaction with chromatin-associated proteins, LANA is essential for cellular persistence of the KSHV episome within the host as it regulates chromatin dynamics and the replication of the KSHV genome with a stable copy number ensuring proper segregation of the episome to daughter cells [102,104,105].

Similar to EBV, the latent KSHV genome is characterized by a specific landscape of epigenetic modifications where viral gene expression is silenced through viral DNA methylation and histone modifications [117]. DNA-hypermethylation was shown to correlate with the suppression of gene expression in PEL cells [118], while inhibition of DNA methyltransferases induces lytic reactivation, highlighting the role of DNA methylation during latency [119]. High levels of repressive histone modifications such as H3K9me3 and H3K27me3 are especially found in late gene regions during latency, as reviewed in Toth et al. [120]. Interestingly, several loci of the KSHV latent genome, such as the immediate early gene replication and transcriptional activator (Rta), still show activating histone modifications such as H3K9/K14ac and H3K4me3, whereas constitutive heterochromatin markers such as H3K9me3 were largely absent. Specific epigenetic patterns identified in the study by Günther and Grundhoff are thought to enable rapid chromatin remodeling for the lytic switch [117].

LANA’s key role in the repression of lytic gene expression is emphasized by its interactions with chromatin-associated and epigenetic regulatory factors, as well as the DNA methylation machinery [121,122]. Moreover, LANA can directly repress lytic reactivation by interacting with DNA-binding proteins such as RBP-jK. This interaction inhibits Rta activation [121,123]. LANA also plays a key role in the oncogenic transformation of KSHV-infected cells; for example, it impairs TGFβ signaling, leading to the inhibition of TGFβ-mediated antiproliferative effects [124]. It was also shown to be involved in JAK3/STAT3 signaling to promote angiogenic factors [125]. Importantly, LANA can directly inhibit tumor suppressor protein p53 and p73 functions, thereby maintaining the viral episome within the infected cell and contributing to oncogenesis [126,127].

In contrast to latency, the repertoire of viral lytic genes is much bigger and can be activated by host and viral factors as well as environmental stimuli in a strictly regulated manner from immediate early genes required for gene transcription such as ORF50 (Rta), ORF45, K8α, K4.2, K4.1, K4, ORF48, ORF29b, K3, and ORF70, to early genes involved in DNA replication (K8, K5, K2, K12, ORF6, ORF57, ORF74, K9, ORF59, K3, ORF37, K1, K8.1, ORF21, vIL-6, PAN RNA, vIRF1, K1, and ORF65), to late genes like glycoproteins gB and K8.1 involved in viral assembly [125,128,129,130]. Of these, Rta, encoding for an E3 ubiquitin ligase involved in dynamic chromatin remodeling, is a critical lytic gene. Once active, it leads to the degradation of viral lytic replication repressors and the activation of lytic promoters, resulting in lytic gene expression and the production of new infectious virions. Rta can initiate DNA binding of RBP-jK, thereby activating the Notch signaling pathway, which was shown to be sufficient to activate lytic replication [131,132,133]. Epigenetic modifications caused by overexpression of histone demethylases or inhibition of transcriptional repressors such as PcG proteins were also found to be critical for the switch between latency and lytic replication [134]. Since KSHV proteins interact with the host cell chromatin, it is plausible that epigenetic reprogramming plays an important role in the pathogenesis of KSHV-associated malignancies [135].

The KSHV lytic phase can be reactivated by inflammatory cytokines such as hepatocyte growth factor (HGF), oncostatin M (OSM), or IFNγ that are also stimulated by other infectious agents [108]. Various co-infections besides HIV, such as HSV-1 and -2, HHV-6, and HCMV are potent cofactors that activate KSHV lytic reactivation and thereby affect KSHV pathogenesis, highlighting the complexity of mechanisms and the influence of co-infections for disease development and outcome [136,137,138], see Section 4.

Once activated, several lytic genes such as viral protein kinase (vPK/ORF36), viral G-protein-coupled receptor (vGPCR/ORF74), viral interferon regulatory factor 1 (vIRF1/K9), and viral IL-6 (vIL-6/K2), as well as membrane proteins K1 and K15, were found to have oncogenic functions, stimulating cellular proliferation, transformation, angiogenesis, and cytokine production [139,140,141,142,143].

## 4. Co-Infection and/or Inflammation Triggers Lytic Reactivation of Oncogenic Gammaherpesviruses

While latency represents a long-term underlying infection with EBV or KSHV of its respective host cells and leads to viral persistence with minimal viral gene transcription, lytic reactivation, potentially triggered by various environmental stimuli, can lead to acute disease onset. The following sections will focus on co-infections relevant in SSA, implicated in EBV- and/or KSHV-associated pathologies.

### 4.1. Lytic Reactivation of EBV

In the context of co-infections, associated inflammation typically leads to the production of cytokines and chemokines and the disruption of signaling pathways, which can promote lytic reactivation. EBV-produced viral cytokine homologs, such as BCRF1 (vIL-10), act as antagonists, which negatively regulate IL-12 resulting in IFNγ production and leading to cellular apoptosis [144]. Upregulation of cytokines such as TNFα, TNFβ, and G-CSF are also suggested to contribute to reactivation [145]. Induction of inflammation by exosomes carrying viral proteins has also been suggested [146]. Different pathways, such as NF-κB, JAK/STAT, MAPK/ERK, and PI3K/AKT pathways, are activated by inflammatory signals and associated with lytic reactivation and subsequent oncogenesis [99].

Co-infections such as with HIV, Mtb, and *Plasmodium* sp. have previously been shown to exacerbate EBV infection, especially in immunocompromised or immunosuppressed individuals, allowing latent EBV infection to opportunistically reactivate [95]. More recently, co-infection with severe acute respiratory syndrome coronavirus 2 (SARS-CoV-2) has been suggested to be associated with lytic EBV reactivation [147].

Serological assays that are able to differentiate between acute or past infections are based on the detection of viral capsid antigen (VCA) IgM and early antigen-diffuse (EA-D) IgG (acute infection) and EBNA-1 IgG (past infection), respectively. However, during lytic reactivation, all EBV antigens are present, i.e., VCA IgM, EA-D IgG, and EBNA-1 IgG [147].

#### 4.1.1. HIV

HIV is an RNA virus from the retrovirus family of lentiviruses [148], being responsible for approximately 84.2 million infections and 40.1 million mortalities worldwide [149]. South Africa has one of the largest HIV-seropositive populations globally [150,151], with approximately 7.5 million people living with HIV, leading to approximately 51,000 HIV-related deaths in 2021 [152]. Although global HIV infection is still rising, the introduction of ART has led to a substantial decrease in AIDS, the most advanced stage of HIV infection, and a higher survival rate for HIV-positive individuals [153,154]. This, on the other hand, has led to a higher prevalence of HIV-related malignancies as the HIV-positive population ages [155,156,157].

HIV-associated lymphoproliferative disorders such as diffuse large B-cell lymphoma (DLBCL), BL, HL, and EBV-associated smooth muscle tumors can be a result of uncontrolled EBV infection due to the immunocompromised state of the host [151,158,159]. A very early study by Rahman et al. (1991) showed EBV lytic reactivation in 49 male HIV seroconverters within 6 months of HIV infection. It was further demonstrated that IgG antibody titers against EBV EA (associated with primary/acute EBV infection) were increased, and EBV infection remained activated even up until 18 months after HIV seroconversion, suggesting a role for this herpesvirus in HIV pathogenesis [160]. Whitehurst et al. (2022) demonstrated a direct effect of HIV co-infection on the pathogenesis and progression of EBV-related diseases. Co-infection with HIV increased systemic EBV replication and immune activation as well as EBV-induced tumorigenesis in a humanized mouse model of EBV infection [161]. EBV/HIV-infected mice produced a greater number of tumors at a distinct anatomical site, with the expression of LMP1 and EBNA-2 indicating latency program III gene expression in all EBV-infected tumors with or without HIV infection [161]. Conversely, a study conducted using HIV-positive samples from a cohort of children demonstrated that EBV co-infection affected HIV viral load: the highest mean HIV viral load was observed in EBV co-infected samples compared to other viral co-infected samples, with an increase in the production of IFNγ, IL-2, and TGFß in the HIV/EBV co-infected group [162,163,164,165]. Importantly, the increased EBV shedding in patients with HIV-1 infection was found to be decreased by the use of ART [166,167,168].

It is suggested that the CD4^+^ T-cell depletion and associated CD8^+^ T-cell deterioration and NK cell differentiation to the less protective CD56^−^CD16^+^ phenotype due to HIV infection increases EBV-associated malignancies [169]. However, it has also been shown that B cells are more susceptible to HIV infection during EBV B-cell transformation in a CXCR4- and CD4-dependent manner in vitro [170], suggesting a bidirectional effect of HIV/EBV co-infection.

#### 4.1.2. Mycobacterium Tuberculosis

The acid-fast bacterium Mtb is the causative agent of TB, affecting approximately 10.6 million individuals worldwide, with South Africa contributing approximately 87% of all globally estimated incident TB cases in 2018 [171]. Approximately 53% of HIV-positive individuals in South Africa were reported to be co-infected with Mtb in 2021, representing the most common cause of death among HIV-infected people, accounting for one-third of AIDS-related deaths [152,172].

The direct effect and mechanistic relationship between EBV infection and TB pathogenesis still needs to be evaluated; however, a link due to the high Mtb/HIV co-infection rate in South Africa and the ubiquitous nature of EBV has been suggested, which may be dependent on the stage of EBV infection and timing of Mtb infection [173].

In a very early case report from 1986, EBV was detected in an active TB-positive patient, confirmed by the detection of anti-VCA IgM and IgG antibodies [174]. In a recent case study, it was shown that EBV infection is sometimes overlooked during differential diagnosis, despite the presence of evidence indicating past EBV infection of the patient. This oversight is attributed to the prominence given to the patient’s history of HIV/Mtb infection, which is considered more clinically significant [175].

EBV has been detected in lung TB, which is associated with multidrug resistance (MDR-TB). In a study including 233 patients, EBV was observed in 27% of HIV-negative lung TB patients and in 21.9% of HIV-positive lung TB patients. Although there was no correlation between EBV and the presence or absence of HIV, EBV and the detection of other HHVs were associated with MDR-TB in HIV patients [176].

Interestingly, a murine model of mice latently infected with the murine gammaherpesvirus 68 (MHV68), which is genetically related to EBV and KSHV, demonstrated moderate protection from virulent Mtb infection by limiting Mtb growth and dissemination due to the increased response of IFNγ and the possible elicitation of a polyclonal T-cell response. This suggests that human gammaherpesvirus infection could have a significant influence on TB pathogenesis, although it may often be unrecognized, particularly in countries with a high burden of TB, such as South Africa [177].

#### 4.1.3. *Plasmodium* sp.

Malaria is an infectious disease caused by apicomplexan parasites of the *Plasmodium* genus, which are transmitted by the *Anopheles* mosquito [178]. Malaria accounts for approximately 241 million estimated cases globally, resulting in over half a million deaths in 2020 [178,179]. In SSA, malaria is endemic and accounts for approximately 100 million cases and an estimated 275,000 deaths [178].

Endemic BL, a common pediatric cancer in Africa, is classified by the presence of EBV and *Plasmodium* sp. infection; however, the exact extent of their respective contributions is not yet fully elucidated. It has been reported that the cysteine-rich inter-domain region 1α (CIDR1α) of the *Plasmodium falciparum* membrane protein 1 can directly induce EBV reactivation, thus increasing the risk of BL. CIDR1α activates the memory component of B cells in which EBV is known to persist in a latent form. It was observed that CIDR1α binds to EBV-positive B cells and increases the number of cells that switch to the viral lytic cycle, thereby increasing the immediate early gene BZLF1 (Zta) expression [180].

Co-infection with EBV has been proposed to contribute to a greater risk of malaria-associated morbidity and mortality in children from SSA [181]. The timing of EBV/*Plasmodium* sp. co-infection plays a crucial role in the development of severe malaria, as acute EBV infection during *Plasmodium* infection is essential for the heightened lethality of the co-infection. Since children in SSA often acquire both EBV and *Plasmodium* sp. within their first year of life, it is likely that EBV infection is in the acute phase during their initial malaria episode. This implies that complications leading to childhood malaria deaths may be partially attributed to EBV infection. In experiments conducted on mice, those infected with MHV68 and subsequently infected with *Plasmodium* sp. within 7–15 days experienced severe anemia and succumbed to the infection. However, mice infected with *Plasmodium* at a much later stage did not exhibit the same complications and survived [181].

Evidence of previous *P. falciparum* infection on EBV kinetics was reported by a study on children presenting with acute uncomplicated *P. falciparum* malaria (referred to as acute Kisumu; A-KSM) who showed higher EBV viral loads and stronger anti-VCA IgG antibody responses. This study also looked at KSHV and *Plasmodium* sp. infection suggesting that malaria drives both EBV and KSHV lytic reactivation [182] (see Section 4.2.3).

Other studies have suggested expansion of EBV-infected B cells during malaria, especially in children [37,183]. It has also been shown that binding of *P. falciparum* erythrocyte membrane protein 1 (*Pf*EMP1) to latently EBV-infected B cells caused lytic reactivation with the aid of CIDRα1 indicated by a higher EBV DNA level in the plasma of children and pregnant women with malaria than those without malaria. However, the interplay between host factors in recurrent malaria episodes and co-infection with EBV, leading to the development of endemic BL, still needs to be elucidated [184].

#### 4.1.4. SARS-CoV-2

SARS-CoV-2 is the causative agent of coronavirus disease 2019 (COVID-19), which was declared a global health pandemic in 2020. SARS-CoV-2, a single-stranded RNA virus, is part of the Coronaviridae family and is responsible for approximately 765 million global infections to date [185,186,187,188,189]. The virus infects the upper respiratory tract, with disease progression leading to inflammation of the alveoli in the lungs resulting in limited gaseous exchange, uncontrolled immune responses (also termed the “cytokine storm”), and potentially fatal respiratory distress [190,191].

In South Africa, the SARS-CoV-2 pandemic occurred against the backdrop of high HIV, Mtb, herpesvirus, and respiratory infections, as well as non-communicable disease burdens. The pandemic contributed to approximately 4 million confirmed infections and approximately 100,000 confirmed deaths ranking South Africa as one of the most severely affected countries on the African continent [188,189,192].

Recent studies have shown increasing evidence to suggest a link between SARS-CoV-2 infection and EBV reactivation resulting in heightened fever and inflammatory responses in COVID-19 patients [193]. However, the question of whether SARS-CoV-2 triggers EBV reactivation or if EBV directly exacerbates SARS-CoV-2 infection remains uncertain and requires further investigation.

In a clinical study conducted by Chen et al. (2021) in Wuhan, China, 55.2% of hospitalized COVID-19 patients with confirmed past EBV infection displayed positive EBV VCA IgM, indicative of EBV reactivation within two weeks of testing positive for SARS-CoV-2. This suggested a correlation between EBV reactivation and the acute phase of COVID-19 [193]. Moreover, a study conducted by Paolucci et al. (2020) demonstrated that the overall median EBV DNA level was significantly increased in intensive care unit (ICU) patients compared to sub-intensive care unit (SICU) in Italy, with EBV reactivation observed in 95.2% of ICU patients and 83.6% of SICU patients [194]. The presence of EBV DNA correlated with a significant loss of NK and CD8^+^ T cells in these COVID-19 patients [194]. These data correlate with a study conducted in France demonstrating that EBV reactivation led to longer median ICU stays, with 82% of COVID-19 ICU patients displaying EBV reactivation [194,195]. A retrospective study from Wuhan, China, found that 13.3% of patients in the ICU developed EBV reactivation with higher 28-day and 14-day mortality rates than the remainder of patients without EBV reactivation and who also received more immune-supportive treatment than non-EBV patients [196].

A study by Verma et al. (2021) indicated that the SARS-CoV-2 receptor angiotensin-converting enzyme 2 (ACE2) was upregulated during lytic EBV infection due to the EBV lytic activator, BZLF1 (Zta), targeting methylated promoters, thus increasing SARS-CoV-2 susceptibility in the human oral epithelium [197].

A more recent study suggested that SARS-CoV-2 infection does indeed increase EBV reactivation as demonstrated by the detection of EBV DNA in plasma samples. EBNA-1 IgG indicates previous EBV infection; therefore, primary EBV infections were defined in the study by the absence of EBNA-1 IgG in samples and were excluded from analysis, thus revealing 27.1% of EBV reactivation in the COVID-19-positive sample group, compared to only 12.5% of reactivations in the COVID-19-negative group [147].

To investigate the effect of “long COVID” on EBV reactivation, Gold et al. (2021) reported increased EBV EA IgG or EBV VCA IgM in 66.7% of “long COVID” subjects and in 10% of control subjects with a similar trend observed in patients 21–90 days following acute SARS-CoV-2 infection [198]. This suggests that EBV reactivation may occur soon after or concurrently with acute COVID-19 and as a result of COVID-19-associated inflammation (i.e., the “cytokine storm”) rather than SARS-CoV-2 infection [198]. It has been suggested that EBV reactivation could be the cause of “long COVID” symptoms due to impaired mitochondrial functions during the host immune response to COVID-19 and long-term symptoms [199]. Moreover, the proinflammatory cytokine IL-6 has been associated with increased EBV reactivation in COVID-19 patients, as significant levels of EBV viremia were found in 78% of critically ill COVID-19 patients compared to only 44.4% in non-COVID-19 patients [195]. However, the specific mechanisms underlying this relationship were not clearly defined.

#### 4.1.5. Other Co-Infections

HCMV and EBV co-infection are commonly detected in immune-competent children [200] and is suggested to facilitate multi-pathogen infection such as with respiratory syncytial virus (RSV), Chlamydia pneumonia (CP), HHV-6, -7, and/or measles virus, among others [200]. It is speculated that EBV/HCMV co-infection can influence B-cell responses and NK cell cytokine production and the frequency of NKG2C^+^ and CD57^+^NKG2C^+^ NK cell differentiation; however, the mechanisms governing this are unknown [200,201,202]. Recently, a role for EBV/HCMV-associated reactivation in hematopoietic stem cell transplantation (HSCT) patients has been reported, increasing the incidence of hemorrhagic cystitis and thereby reducing the overall survival rate of HSCT patients [203].

Co-infection of EBV with HSV has been observed in a rare clinical case of Erythema multiforme in a young child. Elevated levels of antibodies to both IgM HSV and EBV were detected with no other viral infections [204]. Another study demonstrated an association of EBV-seropositivity with the presence of HCMV and HSV-1 viral antibodies [205].

It has also been observed that EBV and HPV co-infection leads to a dysregulation of miRNA expression. Both viruses can synergistically dysregulate miR-21 and miR-200c, the expression of which has been shown to have a role in cervical carcinogenesis [206]. EBV was suggested to impair T-cell immunity in advanced stages of cervical cancer, contributing to the immune evasion of HPV-infected epithelial cells. The interaction between EBV LMP1 and HPV viral proteins altered NF-κB and MAPK signaling, among others, in a murine model [207], while another study highlighted the necessity of HPV viral oncoproteins E6 and E7 in EBV lytic replication in oral keratinocytes [208]. A helper role of EBV in cervical cancer development has been suggested by several studies due to the presence of both HPV and EBV DNA sequences in malignant tissues [209,210,211,212,213,214,215,216].

Recently, EBV has been detected in a newly emerging hemorrhagic fever known as thrombocytopenia syndrome (SFTS) caused by a bunyavirus transmitted through tick bites with a fatality rate of up to 30%. EBV reactivation was seen in 4 out of 22 (18.2%) SFTS patients from Qingdao City, China. Previous EBV infection was observed in more than 90% of the SFTS patients in this study as defined by EBV IgG levels suggesting that EBV was reactivated rather than being a primary infection [217].

### 4.2. Lytic Reactivation of KSHV

Factors that are known to induce lytic KSHV reactivation are oxidative stress, hypoxia, inflammation, and immune suppression [108,218], as well as various co-infections such as with HIV, Mtb, *Plasmodium* sp., HSV-1, HSV-2, HHV-6, and HCMV, and more recently SARS-CoV-2 [136,137,138,219,220].

ROS and cytokine production in the context of inflammation and co-infection can induce cellular pathways that involve NF-κB inhibition as well as p38, JNK, and ERK1/2 signaling, ultimately leading to KSHV reactivation [221,222]. Under hypoxic conditions, which usually promote cell cycle arrest and hinder DNA replication in order to decrease the energy consumption of the cell, KSHV LANA has been shown to maintain genome replication of both the host and the virus [223]. Moreover, in vitro studies showed that lytic reactivation occurs upon acute and chronic exposure of PEL B-cell lines to hypoxia, leading to an increase in KSHV lytic proteins as well as secreted vIL-6 [224].

Compared to EBV diagnostics (see Section 4.1), there are no clinically utilized, standardized high-sensitivity serological tests for KSHV infection, and serological differentiation of KSHV acute and past infection, as well as reactivation, is not well established [225]. However, in-house ELISA platforms to quantify lytic K8.1 and latent LANA are well recognized to assess general KSHV seropositivity [226].

#### 4.2.1. HIV

Although the introduction of ART has had a major effect of reducing KS incidence and improving prognosis when included as therapy for HIV-infected individuals, HIV-related immune suppression, e.g., defined by low CD4 counts (<200 cells/µL), still remains the most important mechanism promoting KSHV-driven pathogenesis [59]. While there has been a decline in the incidence of KS in HIV-infected individuals due to ART, a rising number of KS patients with CD4 counts over 300 cells/µL have been reported [227]. This could be explained by HIV-unrelated KS because the life expectancy of HIV-infected patients on ART has increased or to HIV-related effects that do not depend on T-cell immunosuppression. In other KSHV-related diseases, such as MCD, patients show preserved CD4 counts [60], highlighting that KSHV-dependent malignancies are highly heterogeneous, and research in these populations remains a priority in AIDS research.

Lytic reactivation of KSHV in the context of underlying HIV infection was shown to be directly influenced by HIV proteins such as transactivator of transcription (Tat), which can activate the immediate early gene Rta as shown in an in vitro study using a PEL cell line [228]. Moreover, HIV Tat mimics growth factors and signaling molecules and can function as an angiogenic factor, thereby activating the tyrosine kinase receptor VEGF-A through direct binding, resulting in growth of KS lesions [229]. HIV has also been shown to potently induce lytic replication of KSHV via activation of the KSHV Rta in a Tat-independent manner [228].

Late stages of HIV infections and/or AIDS are characterized by chronic inflammation. HIV-1 Tat can induce cytokines, which, in turn, were shown to induce KSHV lytic reactivation [59]. The induction of inflammatory cytokine secretion is due to Tat modulation of signaling pathways controlling the inflammatory response in the cell, such as activation of JAK/STAT [230], contributing to lytic KSHV replication [230]. Lytic replication of KSHV was also found to be induced by soluble factors produced by or in response to HIV-1-infected T cells co-cultured with a KSHV-infected cell line (BCBL-1), as an increase in KSHV mRNA transcripts, proteins, and infectious virions has been shown [231].

Due to HIV-related immune suppression, lytically activated cells are not eliminated by the host immune system, further fueling HIV/KS pathogenesis [232].

#### 4.2.2. Mycobacterium Tuberculosis

The context of the high prevalence of Mtb in SSA has led to overdiagnosis and overtreatment of TB, leaving KSHV-related pathologies that may symptomatically mimic TB misdiagnosed and, therefore, untreated [46]. KSHV-MCD, for instance, is rare but likely underdiagnosed and is characterized by nonspecific inflammation symptomatically similar to HIV/TB, complicating accurate diagnosis. Indeed, when investigating a large patient cohort (*n* = 682) from South Africa presenting with typical TB symptoms, a subset of patients fitted the KICS working case definition [56], with one patient retrospectively being diagnosed with MCD [46].

The impact of Mtb/KSHV co-infection was also addressed in a study comparing KSHV viral loads and KSHV antibodies in patients with underlying Mtb infection, which revealed significantly higher titers in Mtb/KSHV patients than in controls [233]. Virulence factors secreted by Mtb were shown to influence KSHV-infected cells and induce KSHV lytic reactivation in vitro, such as the early secreted antigenic target 6kDa (ESAT-6), which was confirmed by lytic gene expression of Rta, vGPCR, and K8.1 in HUVEC endothelial cells and PEL cells [234]. As ESAT-6 can be secreted into the extracellular space, co-infection between Mtb and KSHV does not have to occur in the same cell, although this is very likely due to similar tropism including macrophages, dendritic cells, epithelial, and endothelial cells [234].

#### 4.2.3. *Plasmodium* sp.

As KSHV seroprevalence is particularly high in SSA, where malaria is endemic, an association between these two diseases has been suggested in the literature [235,236,237]. A study investigating KSHV seroconversion in Kenyan children further showed that infection with *P. falciparum* increases the risk of KSHV seropositivity at an early age [238]. Several factors, such as impaired immune functions, favor the development of KS and other KSHV-associated pathologies. Furthermore, malaria triggers a T helper type 2 (Th2) immune response, which is also very common in helminth infections, and characterized by the production of the cytokines IL-4, IL-5, and IL-13. IL-4 was shown to reactivate gammaherpesviruses in a mouse model and specifically reactivate KSHV in vitro with increased expression of immediate early and late lytic transcripts [239]. A recent study by Olouch et al. (2023) examined KSHV serological patterns of children with acute malaria and demonstrated that KSHV latency was disrupted by acute malaria episodes. Not only was acute *P. falciparum* infection characterized by distinct serological profiles, but KSHV lytic antigens were also increased in patients with acute malaria [182].

Ruocco et al. (2011) proposed an “oncodrug” hypothesis, claiming that cancer development is favored by immunosuppressive properties of anti-malaria drugs such as quinine. Indeed, iatrogenic immunosuppression can last for years, which makes KSHV-infected individuals more susceptible to lytic reactivation [240]. Several species of *Plasmodium* can cause severe anemia [241], which, in turn, triggers hypoxia, a known risk factor for lytic reactivation of KSHV, as stated above [224].

#### 4.2.4. SARS-CoV-2

While several co-morbidities are known to increase the risk of a fatal COVID-19 outcome, co-infection with oncogenic viruses is of particular significance for long-term cancer risk and KSHV-associated pathogenesis [242]. Observational studies conducted in South Africa identified an association between KSHV viral load and COVID-19 outcomes [243]. Although the study design could not determine whether disease synergy was at play, several SARS-CoV-2 proteins have indeed been shown to induce lytic reactivation of KSHV in vitro, such as SARS-CoV-2 Spike (S) and nucleocapsid (N) proteins when ectopically expressed [220]. Not only the virus itself, but drugs that are used in the treatment of COVID-19, such as azithromycin and nafamostat mesylate, but not remdesivir, induced KSHV lytic reactivation in iSKL.219 cells [220]. Further, a potential interplay between SARS-CoV-2 and KSHV was suggested by data showing upregulated ACE2 expression in AIDS-KS tissue, although the underlying mechanisms of KSHV-mediated regulation of ACE2 expression remain unknown [220]. An association between KSHV and SARS-CoV-2 was also indicated in a clinical case of a woman with confirmed KS without visual skin lesions who was hospitalized for COVID-19 and showed subsequent recurrence of KS. Both KSHV and SARS-CoV-2 were detected by transmission electron microscopy (TEM) of the patient’s biopsies, and the recurrence of KS was speculated to be promoted by inflammation caused by SARS-CoV-2 infection [244]. Clinical manifestation of lytic reactivation of other herpesviruses (EBV, HCMV, HSV) was substantially higher in patients co-infected with SARS-CoV-2, especially those with severe symptoms as well as those who were vaccinated against SARS-CoV-2 [245]. Lytic reactivation of KSHV in the post-pandemic era should therefore remain a research focus, especially in populations with a high prevalence of HIV and other infectious diseases.

#### 4.2.5. Other Co-Infections

HCMV, a betaherpesvirus, occurs ubiquitously, can be vertically transmitted, and represents an important opportunistic infection in HIV-infected individuals [59,246,247]. Immunosuppression is critical for the reactivation of both KSHV and HCMV, and interactions between the two viruses are very likely, not least due to similar cellular tropisms. Lytic replication of KSHV was reported to be induced by the direct activation of Rta by gene products of the HCMV UL112-114 locus, encoding four phosphoproteins synthesized via alternative splicing [248]. Latently KSHV-infected human fibroblasts were induced to lytic replication by infection with HCMV, giving rise to new virion progeny. It was further shown that HCMV infection led to higher KSHV replication in endothelial cells and activated lytic replication in keratinocytes [136]. KSHV reactivation by HCMV might be cell type dependent and have different underlying mechanisms, which was suggested by studies in which a laboratory HCMV strain was unable to induce KSHV lytic reactivation in a BCBL-1 cell line [249].

Other herpesviruses, such as HSV-1 and HSV-2, have been described as cofactors with the potential to reactivate KSHV [137,219]. HSV-1 infection was shown to activate Rta through cellular miRNAs. Upon infection with HSV-1, miR-498, and miR-320d (both of which directly target Rta at its 3′ UTR and inhibit its expression) were downregulated. This was corroborated in experiments overexpressing miR-498 and miR-320d, which resulted in the inhibition of lytic reactivation, whereas miR-498 and miR-320d repression enhanced lytic replication of KSHV [219]. In addition, the PI3K/AKT and ERK/MAPK pathways were found to be stimulated by HSV-1 infection-promoting lytic reactivation [250]. Likewise, HSV-2 infection was shown to lead to Rta activation, resulting in lytic KSHV gene transcription, expression of viral proteins, and production of infectious viral particles in BCBL-1 [137]. Furthermore, HHV-6 was found to be a potent factor inducing lytic replication of KSHV. Co-culture experiments of BCBL-1 cells with HHV-6-infected cells showed an increase in KSHV lytic transcripts after 8 h, which further induced other proteins of the lytic cycle [138]. HHV-6 could also influence the KSHV lytic cycle by regulating cytokines such as TNFα, IL-β, and IFNs [251].

### 4.3. The Interplay between EBV and KSHV

Co-infection with the two gammaherpesviruses, EBV and KSHV, has been described in immunocompromised hosts with AIDS-associated PEL. Both viruses negatively affect the others’ gene expression level following physical interaction on a molecular level. EBV lytic replication is inhibited by KSHV Rta, while KSHV lytic gene expression is inhibited by EBV BZLF1 (Zta) [252].

KSHV lytic replication can also be suppressed by EBV LMP-1. However, LMP-1 can be induced by KSHV Rta in latently EBV-infected cells, which can result in the suppression of lytic replication of both EBV and KSHV. Although EBV and KSHV can suppress each other’s lytic replicative cycles during co-infection, they can be mutually beneficial during long-term latency, whereby they enable each other to escape the host’s immune response [59].

Interestingly, KSHV Rta has also been suggested to increase EBV infectivity by upregulating the expression of the EBV entry receptor CD21 [59]. Another study demonstrated that during dual infection of EBV and KSHV, EBV’s ability to transform B cells in vivo was adequate to facilitate KSHV infection [36], which suggests that EBV may also enhance KSHV genome maintenance [33].

In vitro studies further demonstrated that both viruses could be maintained in a BC-1 cell line, derived from a body cavity-based B-cell lymphoma, and could be differentially induced from their latent state into their lytic state using sodium butyrate and 12-O-tetradecanoylphorbol-13-acetate (TPA), respectively. However, only EBV displayed the release of infectious virions from BC-1 cells, whereas there was little evidence that KSHV intranuclear nucleocapsids were released [253]. This shows that the mechanistic pathways for lytic reactivation of these two viruses could slightly differ, allowing for differential induction.

## 5. Conclusions

The high prevalence of infectious diseases in SSA, primarily HIV/AIDS, TB, and malaria represent an unknown risk for pathologies associated with the oncogenic gammaherpesviruses, EBV and KSHV. Lytic reactivation of these viruses is often associated with nonspecific inflammatory symptoms and is, therefore, likely to be misdiagnosed as a pathology caused by one of the more predominant infectious agents in the SSA context. More recently, SARS-CoV-2 infection might impact EBV/KSHV-associated diseases, which could, in turn, exacerbate COVID-19 progression and outcome and/or lead to “long COVID” and/or malignant transformation.

There are neither prophylactic nor therapeutic vaccines against EBV and/or KSHV infection available; therefore, clinical awareness of EBV- and KSHV-associated pathologies and early diagnosis of diseases linked to these two human oncogenic herpesviruses will facilitate successful intervention strategies, particularly in vulnerable populations with high HIV prevalence and high exposure to other circulating infectious diseases. Identification of these populations and implementation of diagnostic platforms for EBV and KSHV in relevant clinical settings is highly recommended. Moreover, anti-oxidant and anti-inflammatory drugs for the symptomatic treatment of co-infections could be promising preventative and therapeutic means for effectively targeting oncogenic herpesvirus reactivation (Figure 1).

## Figures and Tables

**Figure 1 ijms-24-13066-f001:**
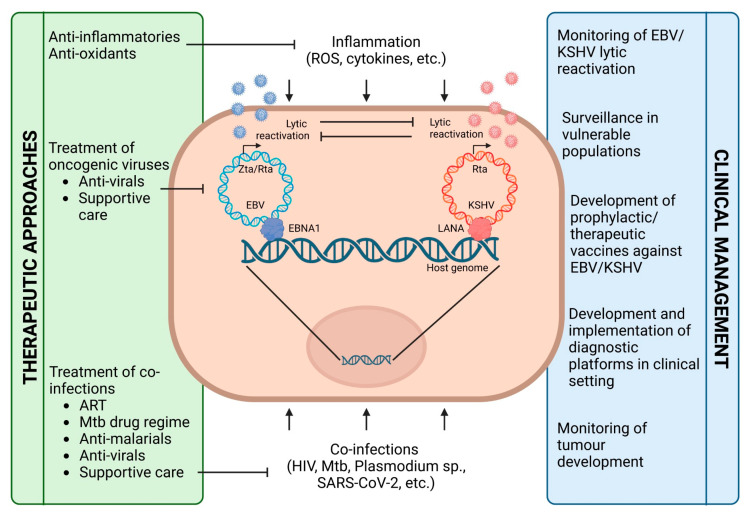
Co-infections and inflammation impact EBV and KSHV lytic reactivation. Preventative and/or therapeutic approaches using anti-inflammatory and anti-oxidative drugs, as well as specific treatment of co-infections, may improve EBV/KSHV-associated disease outcomes, which should be monitored in high-risk populations. (Figure created with Biorender.com (accessed on 14 July 2023)).

## Data Availability

The data that support the findings of this article are openly available at PubMed (https://pubmed.ncbi.nlm.nih.gov/ (accessed on 18 August 2023)).

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
