# Peer review of "The Impact of Co-Infections for Human Gammaherpesvirus Infection and Associated Pathologies"

_ijms, 2023, doi:10.3390/ijms241713066_

Round 1
Reviewer 1 Report
The authors have expressed the view that HIV, TB, malaria, or Covid-19 pathologies mask clinical symptoms that could simultaneously occur during a reactive or primary EBV and KSHV infection. It would aid the reader if the authors included a short paragraph outline primary EBV infection occurring in pediatric or infectious mononucleosis as well as clinical symptoms during primary KSHV.
The authors emphasize the role of B cells in EBV infection however primary EBV infection, reactivation and virus production also occur in oral epithelial cells. A short paragraph or sentence alluding to this known fact would be appropriate and broaden the clinical scope of their SSA co-infective agents.
Author Response
Reviewer 1:
1. The authors have expressed the view that HIV, TB, malaria, or Covid-19 pathologies mask clinical symptoms that could simultaneously occur during a reactive or primary EBV and KSHV infection. It would aid the reader if the authors included a short paragraph outline primary EBV infection occurring in pediatric or infectious mononucleosis as well as clinical symptoms during primary KSHV.
Author reply: We have added the following information (lines 99-104): “Studies have established that most primary EBV infections in children under 2 years of age are asymptomatic; however, some children between 10-48 months old can display symptoms such as fever, tonsillar pharyngitis, prominent cervical lymphadenopathy, and respiratory symptoms during IM [23-25] with a large difference in heterophile antibody responses between infants less than 2 years of age (27.3%), children between 25 and 48 months old (76.2%) and young adults (96%) [24].”
and (lines 156-158): “… due to similar clinical presentation of lytic KSHV infection (such as fever, night sweats, inflammation and respiratory symptoms)…”
2. The authors emphasize the role of B cells in EBV infection however primary EBV infection, reactivation and virus production also occur in oral epithelial cells. A short paragraph or sentence alluding to this known fact would be appropriate and broaden the clinical scope of their SSA co-infective agents.
Author reply: We have added the following information (lines 171-176): “EBV causes primary infection by replicating in the mucosal epithelium, thereby initially infecting oropharyngeal mucosa cells prior to infection of resting B lymphocytes in the underlying secondary lymphoid tissues [67]. The EBV virions generated and released from the epithelial cells favour B cells due to the presence of particular envelope glycoproteins and vice-versa, thereby also allowing the virions to particularly infect naïve B lymphocytes [68,69].”

Reviewer 2 Report
The review summarized a large quantity of information about the effect of co-infections in the regulation of the human gammaherpesviruses EBV and KSHV. While I found the review very useful for the EBV and KSHV field, there are several inaccuracies, wrong references, or missing references in the text. I suggest that the authors check every reference throughout the review and try to cite original papers rather than other reviews. Also, Figure 1 would fit better at the beginning of the review than at the very end.
Comments:
Add a reference to the sentence in line 141-144.
Check the sentence in line 145-147. The end of the sentence is unclear.
Check the references to make sure that the correct citations were linked to the statements/sentences throughout the review.
E.g., line 140-ref 51 does not seem to fit to the statement.
Line 152: “in” is missing before MCD.
Line 173: “These viral proteins and RNAs…”
Line 183-185. Rephrase the sentence because it is misleading. EBNA-1 is the EBV latent protein that tethers the viral genome to the host chromosomes and not other latent factors induced by EBNA-1. Also, ref 66 is not the best fit here, which talks about lytic reactivation.
Line 200: uncontrolled replication of EBV or the latently infected cells? Please specify it.
Line 201: “…viral gene expression occurs in a…”
Reference is missing at the sentence written in line 210-212.
Line 237: what does the 1% refer to?
Line 255-256: the high levels of repressive histone marks were found on the KSHV genome. Please specify it in the sentence.
Line 256-258. The statement is inaccurate. High level of H3K9me3 can be found on specific regions of the KSHV genome in PEL cells (PMID: 20661424, 20532208, 21228229).
Line 271-277: there are no references. There aren’t that many IE genes as listed in the sentence. Where does this information come from. There is no K8.2 in KSHV.
Line 320. Reference is missing for SARS-CoV-2induced EBV reactivation.
Line 339. What is EA IgG?
Line 464-467. The statement is inaccurate. Detection of EBNA-1 shows latent/persistent infection and not primary infection. Please check the paper and rephrase the sentence.
Line 518-520. LANA maintain genome replication of the host? Please rephrase the sentence and add reference to the sentence.
Line 555. BCBL-1 and not BVBL-1
English is fine.
Author Response
Reviewer 2:
1. The review summarized a large quantity of information about the effect of co-infections in the regulation of the human gammaherpesviruses EBV and KSHV. While I found the review very useful for the EBV and KSHV field, there are several inaccuracies, wrong references, or missing references in the text. I suggest that the authors check every reference throughout the review and try to cite original papers rather than other reviews. Also, Figure 1 would fit better at the beginning of the review than at the very end.
Author reply: We have carefully revised the manuscript and corrected/added all references where necessary. However, we feel that Figure 1 is more relevant at the end of the text as it serves as a summary of the preceding text.
2. Add a reference to the sentence in line 141-144.
Author reply: References have been added (now lines 150-153).
3. Check the sentence in line 145-147. The end of the sentence is unclear.
Author reply: This sentence reads (now lines 154-155): “In addition to HIV, co-infection such as with Mtb, Plasmodium sp., HCMV and HSV is highly prevalent in SSA.” All abbreviations have been written out earlier in the text, and we are not sure what is unclear about this sentence. We have now replaced the word “prevalent” with “common” in the hope that this makes this sentence clear.
4. Check the references to make sure that the correct citations were linked to the statements/sentences throughout the review, e.g., line 140-ref 51 does not seem to fit to the statement.
Author reply: We have carefully checked all references and updated/corrected where necessary, including the one in line 140 (now line 150, ref 59).
5. Line 152: “in” is missing before MCD.
Author reply: This has been corrected (now line 163).
6. Line 173: “These viral proteins and RNAs…”
Author reply: This has been corrected (now line 190).
7. Line 183-185. Rephrase the sentence because it is misleading. EBNA-1 is the EBV latent protein that tethers the viral genome to the host chromosomes and not other latent factors induced by EBNA-1. Also, ref 66 is not the best fit here, which talks about lytic reactivation.
Author reply: We have corrected this section which now reads (now lines 198-202): “EBNA-1 is a sequence-specific multifunctional DNA-binding protein responsible for EBV episomal stability, persistence, and maintenance during latency 0 due to its anti-apoptotic properties and its ability to act as a transcriptional transactivator such as for LMP-1 [86-88]. EBNA-1 tethers the viral episome to the host genome and leads to subsequent B cell immortalization [87].”
8. Line 200: uncontrolled replication of EBV or the latently infected cells? Please specify it. …
Author reply: This has been corrected: “…uncontrolled replication of EBV…” (now line 217).
9. Line 201: “…viral gene expression occurs in a…”
Author reply: This has been corrected (now line 218).
10. Reference is missing at the sentence written in line 210-212.
Author reply: Reference has been added (now line 229).
11. Line 237: what does the 1% refer to?
Author reply: We have revised this sentence which now reads (now lines 253-255): “Whereas 99% of tumor cells (primarily of endothelial origin) in KS lesions are latently KSHV-infected, high rates of proliferation occur in so-called spindle cells accounting for only 1% of tumor cells…”
12. Line 255-256: the high levels of repressive histone marks were found on the KSHV genome. Please specify it in the sentence.
13. Line 256-258. The statement is inaccurate. High level of H3K9me3 can be found on specific regions of the KSHV genome in PEL cells (PMID: 20661424, 20532208, 21228229).
Author reply (comment 12 and 13): This paragraph has been corrected and rephrased (now lines 274-280): “High levels of repressive histone modifications such as H3K9me3 and H3K27me3 are especially found in late gene regions during latency, as reviewed in Toth et al. [121]. Interestingly, several loci of the KSHV latent genome, such as the immediate early gene replication and transcriptional activator (Rta) still show activating histone modifications such as H3K9/K14ac and H3K4me3 whereas constitutive heterochromatin markers such as H3K9me3 were largely absent. Specific epigenetic patterns identified in the study by Günther and Grundhoff are thought to enable a rapid chromatin remodelling for the lytic switch [118].”
14. Line 271-277: there are no references. There aren’t that many IE genes as listed in the sentence. Where does this information come from. There is no K8.2 in KSHV.
Author reply: We have taken out K8.2 and checked that the listed IE genes were correct. We have added the relevant references (now line 298).
15. Line 320. Reference is missing for SARS-CoV-2 induced EBV reactivation.
Author reply: Reference has been added (now line 343).
16. Line 339. What is EA IgG?
Author reply: The sentence has been revised as follows (now lines 363-364): “It was further demonstrated that IgG antibody titres against EBV-EA (associated with primary/acute EBV infection)…”
17. Line 464-467. The statement is inaccurate. Detection of EBNA-1 shows latent/persistent infection and not primary infection. Please check the paper and rephrase the sentence.
Author reply: This has been corrected as follows (now lines 492-496): “EBNA-1 IgG indicates previous EBV infection; therefore, primary EBV infections were defined in the study by the absence of EBNA-1 IgG in samples and were excluded from analysis, thus revealing 27.1% of EBV reactivation in the COVID-19-positive sample group, compared to only 12.5% of reactivations in the COVID-19-negative group [148].”
18. Line 518-520. LANA maintain genome replication of the host? Please rephrase the sentence and add reference to the sentence.
Author reply: This has been rephrased as follows (now lines 549-552): “Under hypoxic conditions, which usually promote cell cycle arrest and hinder DNA replication in order to decrease energy consumption of the cell, KSHV LANA has been shown to maintain genome replication of both the host and the virus [227].”
19. Line 555. BCBL-1 and not BVBL-1
Author reply: This has been corrected (now line 587).
